# Advancing One Health Principles in Bulgaria: A Mixed-Methods Approach to Transdisciplinary Research and Practice

**DOI:** 10.3390/ijerph22040469

**Published:** 2025-03-21

**Authors:** Desislava Vankova, Petya Boncheva, Silviya Mihaylova, Zhaneta Radkova, Ilina Micheva

**Affiliations:** 1Faculty of Public Health, Medical University of Varna, 9000 Varna, Bulgaria; petya.boncheva@mu-varna.bg; 2Medical College, Medical University of Varna, 9000 Varna, Bulgaria; silviya.mihaylova@mu-varna.bg; 3University Publishing Department, Medical University of Varna, 9000 Varna, Bulgaria; radkova@mu-varna.bg; 4Faculty of Medicine, Medical University of Varna, 9000 Varna, Bulgaria; ilina.micheva@mu-varna.bg

**Keywords:** public health, One Health, Bulgaria, Medical University of Varna, antimicrobial resistance, wellbeing, education

## Abstract

The One Health (OH) concept emphasizes public health collaboration and transdisciplinary research. Following the COVID-19 crisis, Bulgaria’s first OH scientific initiative was launched at the Medical University of Varna to raise national awareness and promote OH principles. Expanding on WHO’s holistic OH definition, the project explores “One Health—Integrated Social and Scientometric Approaches for Better Quality of Life”, involving researchers, students, and the University Publishing Department. This report aims to outline the project’s parallel mixed-methods design (PMMD) and initial findings. Using the nominal group technique and horizon scanning, the project was structured into three thematic research areas: (1) OH and antimicrobial resistance (AMR)—assessing knowledge, attitudes, and practices (KAP) among medical and veterinary professionals, and investigating the potential of integrative approaches to manage AMR; (2) OH and wellbeing—examining the wellbeing levels among students and among chronically ill patients; and (3) OH education and dissemination—integrating OH into education and public awareness campaigns. Preliminary findings show policy gaps, particularly the lack of a national OH Action Plan. A KAP survey was conducted (sample size—228, 2023–2024) and wellbeing studies are ongoing (193 students participated, Februarty 2025), while OH topics are being integrated into the curricula. Despite low awareness in Bulgaria, this project seeks to attract national attention to the EU OH strategies.

## 1. Introduction

One Health (OH) is a holistic idea which seeks the harmonious planetary equilibrium between humans, animals, plants, and the living environment for balanced and healthy ecosystems [1,2]. The importance of OH has grown globally, particularly in tackling the public health challenges such as antimicrobial resistance (AMR), zoonotic diseases, and the intersection between environmental and human health. Strengthening public health collaboration and promoting transdisciplinary research are foundational principles of the OH concept [3]. The COVID-19 pandemic further underscored the relevance of OH, demonstrating its unifying social power and interdisciplinary research potential by bridging the gap between the humanities and medical sciences.

Against this background, Bulgaria faces significant AMR-related challenges, with some of the highest levels of antimicrobial use in the European Union. This poses a severe public health risk with cross-border implications, necessitating OH approaches and global public health attention [4]. Despite the pressing need for OH strategies, national-level OH initiatives have been limited. Recognizing this gap, the first national OH scientific initiative was launched in late 2022 at the Medical University of Varna (National Scientific Registry/NACID № 2412381) [5]. This initiative aligns with broader international efforts to implement OH principles and aims to address Bulgaria’s specific public health concerns, particularly in relation to AMR and infection prevention and control (IPC).

The primary ambition of this academic research project was to raise awareness about OH by translating and promoting OH principles at the national level. Building on prior research and fieldwork, and in alignment with the holistic WHO definition of OH [2], the project has expanded to a broader theme: “One Health—Integrated Social and Scientometric Approaches for Better Quality of Life”. This initiative actively engages experienced researchers, PhD candidates, students, and the University Publishing Department at the Medical University of Varna. A distinctive feature of this project is its multidimensional approach—not only does it address classic OH areas such as AMR and IPC, but it also explores integrative perspectives on AMR [6], particularly in relation to antibiotic misuse and overuse.

Moreover, the project extends its focus beyond AMR to socially significant issues such as the subjective health of chronically ill patients and the wellbeing of healthcare professionals. Although these research fields may initially appear diverse, the OH framework serves as an overarching paradigm that unites them under a shared commitment to scientific advancements for societal benefit.

This report aims to achieve two primary objectives:To present the parallel mixed-methods design (PMMD) of the OH academic research project at the Medical University of Varna. This methodological framework ensures a comprehensive approach to investigating OH-related public health challenges.To outline the project’s initial educational steps and interim quantitative research findings, highlighting its contribution to public health initiatives at both the national and EU levels.

By integrating scientific, educational, and social perspectives, this project offers a novel contribution to the field of OH in Bulgaria. Unlike previous OH initiatives, it applies a structured research framework to not only identify but also actively address national OH challenges through education, policy recommendations, and interdisciplinary collaboration.

## 2. Materials and Methods

### 2.1. Study Design and Sampling

The development of the PMMD was based on the nominal group technique, which facilitated brainstorming among a team of experts from human and veterinary medicine, public health, and academia [6]. The process followed the standard steps of independent idea generation, group discussion, and prioritization through ranking. Further, a horizon-scanning procedure and content analyses were employed to identify policy, research, and educational gaps related to OH at the national level. Finally, the project was structured into three main thematic research areas, called chapters (Figure 1):

### 2.2. Chapter “One Health and AMR”—Revealing the Potential of the Classic and Non-Conventional OH Approaches to Manage AMR

-A cross-sectional anonymous self-administered survey (paper version) took place from May 2023 to August 2024, assessing knowledge, attitudes, and practices (KAP) regarding IPC, healthcare-associated infections (HAI), and AMR among medical and veterinary professionals and health administrators, as well as knowledge of the OH concept. Further, semi-structured interviews with experts were planned as a follow-up according to the analysis of the results. The ethics committee approval preceding the survey included the volunteer’s informed consent form and the consent of the four hospitals’ managers, where the survey was carried out. Formal permission was preliminarily acquired by the Bulgarian Food Safety Agency (BFSA) for the distribution of the specifically designed instrument among veterinarians. The research instruments are briefly described below.-Research with a focus on the potential of integrative approaches to manage AMR, including phytotherapy and lifestyle determinants (culture, consumers’ behaviours, and lifestyle determinants of AMR) [7].

### 2.3. Chapter “One Health and Wellbeing”—Focusing on OH’s Broader Mission to Foster the Wellbeing of Society and Consumers/Patients in Healthcare and to Support Professional Capacity Building in the Public Health Sector

-A sociological survey on the wellbeing levels among nursing (online version) and postgraduate (paper version) students at the Faculty of Public Health and the Medical College of the Medical University of Varna. The survey was anonymous and self-administered. The research instruments are briefly described below.-Assessing the importance of measuring subjective health in chronically ill hematological patients. Two instruments were designed to target these patients and their physicians. These instruments were developed at the beginning of 2024, received additional ethical approval, and were a part of a Master’s thesis, the results of which have been published elsewhere [8].

### 2.4. Chapter “One Health—Education and Dissemination”—Bringing OH to the Top of Future Professionals’ and Peoples’ Agenda

-Educational and public awareness campaigns—as a novel concept, OH is not part of the standard curriculum in Bulgarian universities, leaving healthcare professionals ill-equipped to apply its principles. Educating the students but also informing the public about the OH approach and its benefits could garner support for its adoption and implementation. The open access nature and broad dissemination capacity of the University Publishing Department are essential in this chapter.-Academic education and research (including the training of Master’s and PhD students) to support policy integration of OH and its impact on the wellbeing of the Bulgarian society.

### 2.5. The Instruments and Data Analysis

The following instruments have been embedded in the wider mixed-methods design: the KAP questionnaires in the Chapter “One Health and AMR” and the Happiness and Health questionnaire in the Chapter “One Health and Wellbeing”.

#### 2.5.1. The KAP Questionnaires

The self-administered instruments (paper versions) were designed to assess KAP related to IPC measures, AMR and HAI, and knowledge of OH approaches.

Two tools (structured questionnaires) were developed with an identical structure for comparison reasons but specifically created for the following targeted groups: medical doctors and healthcare personnel; veterinarians and experts from the BFSA. Two types of questions were mixed, and the specific quantitatively measurable questions included open qualitative narrative possibilities. Copies of the instruments in Bulgarian are available upon request from the corresponding authors. The respondents did not receive any incentive to fill in the questionnaire.

#### 2.5.2. The Happiness and Health Questionnaires

The self-administered instruments (online and paper versions) were designed to assess subjective information on wellbeing and health among nursing and postgraduate students (kindergarten caregivers, nurses, and pedagogues who participated in the qualification courses at the Medical University of Varna).

Two tools (structured questionnaires) were developed, and the distribution mode was adapted to the specific groups, with an online version for the nursing students and a paper version for the postgraduate students.

The online version contained 11 questions focused on the following areas: subjective assessment of happiness and what determines their happiness; subjective assessment of their health and the level of awareness about factors affecting health; alcohol use and demographic questions. The questionnaire was distributed online, as a Google form, from where the answers were automatically received at the research team’s email address. The questions were multiple-choice, with one open-ended question at the end to freely express their comments related to the discussed topic. Participation in the study was voluntary and anonymous, after obtaining an informed consent form signed by the respondents.

The paper version designed for postgraduate students contained 17 questions, organized into the following areas: self-assessment of personal happiness and what makes them happy; self-assessment of health and its determinants; self-assessment of stress levels; level of life-satisfaction; demographic data—gender, age, and socioeconomic information—education, income. The questionnaires were printed on paper and distributed by the principal investigator. The questions were multiple-choice, with an opportunity at the end for respondents to freely express their comments related to the discussed topic. Participation in the study was voluntary and anonymous, after obtaining an informed consent form signed by the respondents.

Descriptive statistical methods were applied. The IBM SPSS Statistics package, version 25, was used for the statistical processing of primary information. The analysis of the data is at the statistical processing stage, including verification of the information and coding of the results.

## 3. Interim Results and Initial OH-Promotion Activities

The findings are framed according to the predefined chapters. Therefore, the interim results and analyses are reported in a similar manner. Most of the quantitative and qualitative results will be published elsewhere as part of the planned PhD studies.

### 3.1. Chapter “One Health and Antimicrobial Resistance”—Searching for Ways to Manage a Critical Public Health National Challenge with a Negative Cross-Border Impact

The content analysis of national policy documents identified significant gaps, most notably the lack of a comprehensive OH Action Plan. In the National Health Strategy (Decision No. 662 of the Ministerial Council from 29 September 2023) the term One Health (translated as “Edno Zdrave”) is mentioned five times, both in the introductory words of the Minister of Health, and in the main text in Policy 3.7, the aim of which is “limiting AMR in the healthcare system, taking into account the WHO concept of “One Health” and the interconnected problems in human, veterinary medicine and the environment”. The Strategy [9] recognizes that “the effective implementation of Policy 3.7 is possible only through integrated engagement through the “One Health” approaches of the Ministry of Health, the Ministry of Agriculture, the Ministry of Environment and Water, as well as the leading institutions such as the National Centre for Communicable and Parasitic Diseases (NCIPD), BFSA, academic institutions, etc.”.

The survey targeting medical and veterinary professionals was conducted to identify gaps in IPC practices related to AMR and OH-related knowledge. The characteristics of the two groups—medical doctors and veterinarians—are summarized in Table 1 (*n* = 228, >18 years old). The size of the sample was determined by the period (May 2023 to August 2024) and the respondents could participate only once. Overall, 228 volunteers took part, 168 medical professionals and 60 veterinarians. The size and sociodemographic features of the target groups allow for comparisons and correspond with the gender and age characteristics of the professionals’ communities in the country. The rest of the data analysis is the subject of a further study, and a PhD project has been initiated, framing the IPC control investigation within the OH concept.

The literature reviews on consumers’ behaviours and lifestyle determinants of AMR at a national level resulted in two publications in national journals. Further, an idea was conceived and now a PhD has been started.

Regarding the research on the potential of integrative approaches to manage AMR, a monograph on the integrative medicine concept and its educational and research approaches has been published. Focusing on phytotherapy as an integrative therapeutic method to delay antibiotic prescriptions, an article has been published as a result of a desk review on the role of AI in the detection of antimicrobial effects in bioactive phytocompounds.

### 3.2. Chapter “One Health and Wellbeing”—Focusing on OH’s Broader Mission to Promote Societal Wellbeing

The survey assessing wellbeing and health levels among bachelor’s degree students (online version) covered nursing students at the University in 2023 and 2024 (winter semester). The mean age of the sample was 29 years (SD ± 9.5); up to February 2025, the sample size included 308 volunteers, and the response rate was 76% for the graduate students and 90% for the postgraduate students).

The paper-based survey assessing the wellbeing and health levels of postgraduate students included kindergarten caregivers, nurses, and pedagogues enrolled in qualification courses at the Medical University of Varna in 2023 and 2024. The mean age of the sample was 51 years of age (SD ± 12) and up to February 2025 the sample size included 193 volunteers. The summarized sociodemographic characteristics are presented in Table 2.

This part of the project is ongoing. Therefore, the sample is still growing, and the analyses will be performed later during the next year.

### 3.3. Chapter “One Health—Education and Dissemination”—Bringing OH to the Top of Future Professionals’ and People’s Agenda

Educational and public awareness campaigns have been organised to introduce the novel concept of OH to our students. With the belief that educating the students while also informing the public about the benefits of the OH approaches could garner support for its implementation, the following activities were registered during the 5th and 6th European Public Health Week (EPHW), held in May 2023, 2024 (References 24 May 2023—HW66 “Celebrating One Health and Education on the Day of the Bulgarian Enlightenment”, 15 of May—HW12 “Introducing the One Health Concept in the Bulgarian Medical and Public Health Curriculum. Global Educational Approaches”) [10,11,12].

Initially, one master’s and one PhD student were invited to participate in the project. Over time, the master’s thesis evolved into a PhD project, and an additional PhD student was inspired by the OH concepts.

## 4. Discussion

This brief discussion is devoted to the acceptance of the novel OH idea in Bulgaria. The presented design and interim results aim to be one of our wake-up calls for the broad spectrum of stakeholders, because OH represents a promising framework for tackling critical challenges like AMR at a national level.

In Bulgaria, innovations often take time to gain traction, likely due to the prevailing conservatism in maintaining the status quo [13,14]. Resistance to adoption may also stem from misunderstandings, as literal translations from the Anglo-Saxon literature often fail to resonate as powerfully in native Bulgarian. Many foreign terms in the lexicon of public health share this fate. A notable example is the term *One Health*, which is officially translated as *Edno Zdrave* (One Health) in policy documents. However, to better convey the philosophical essence of the movement, the translation *Edinno Zdrave* (Unified Health) has been proposed and disseminated through publications. Through the reported project, the Medical University of Varna has proven to be an incubator of new ideas and an early adopter [15] of novel approaches such as OH at a national level.

## 5. Conclusions

At present, OH remains a largely unfamiliar and unimplemented concept in Bulgaria. In contrast, the European Union recommends OH strategies as a critical approach to addressing complex public health challenges, such as AMR, raising costs, and environmental pollution. Over the course of the post-COVID-19 period, the OH framework has established itself as both a key policy term demanding political attention and a “tool for action” for the EU community.

The presented national project aspires to advance OH principles in Bulgaria by drawing political, policy, and academic attention to the concept. Translating OH ideas into practical application will require promoting OH-focused training to foster a shared understanding of the interconnections between all the societal domains. Building on this study’s findings, future research should further explore the adoption and implementation of the OH concept in Bulgaria and similar sociocultural contexts. Integrating OH into Bulgarian medical curricula and policy frameworks could improve national AMR strategies.

## 6. Strengths and Limitations of the Study

This study integrates quantitative surveys and qualitative insights, enhancing validity, while ethical approvals strengthen reliability. The study contributes to OH education, policy, and public awareness in a country that has been an EU Member State since 2017.

However, limitations of this study include limited generalizability, potential response biases, and the interim nature of the findings. Additionally, linguistic and cultural adaptation challenges may impact OH acceptance. Despite these constraints, the study provides crucial early insights, supporting further research and policy integration of OH in Bulgaria.

## Figures and Tables

**Figure 1 ijerph-22-00469-f001:**
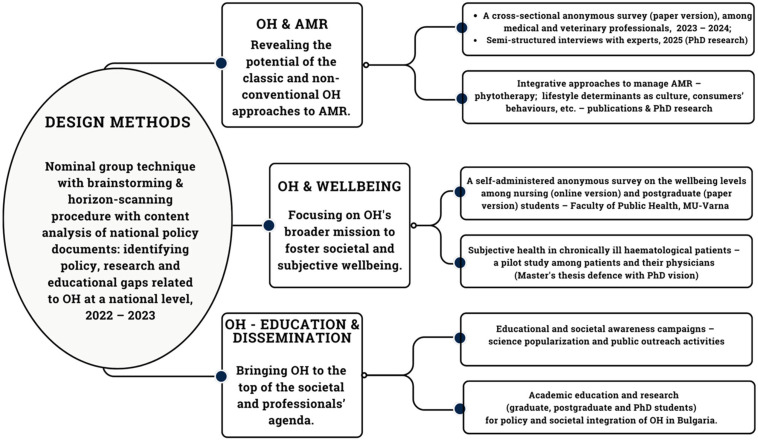
The parallel mixed-methods design (PMMD). Topic: ***One Health—Integrated Social and Scientometric Approaches for Better Quality of Life*** (Project № 22007, Science Fund, Medical University of Varna). Authorship: the figure was created by the Authors.

**Table 1 ijerph-22-00469-t001:** Sociodemographic characteristics of the professionals sample.

Respondents	Frequency N	Male N (%)	Female N (%)	Age Range	Median Age (SD)
Medical professionals	168	16 (9.5%)	152 (90.5%)	26–69	49 (±12.07)
Veterinary professionals	60	28 (46.7%)	32 (53.3%)	29–65	44 (±16.1)
Total	228	44 (19.3%)	184 (80.7%)	26–69	47 (±14)

**Table 2 ijerph-22-00469-t002:** Sociodemographic characteristics of the students sample.

Respondents	Frequency N	Male N (%)	Female N (%)	Age Range	Median Age (SD)
Students	308	9 (3%)	299 (97%)	18–56	29 (±9.5)
Postgraduate students	193	0	193 (100%)	28–72	51 (±12)

## Data Availability

The instruments and the dataset used and analyzed during the current study are available from the corresponding author on request.

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
