# Peer review of "Advancing One Health Principles in Bulgaria: A Mixed-Methods Approach to Transdisciplinary Research and Practice"

_ijerph, 2025, doi:10.3390/ijerph22040469_

Round 1
Reviewer 1 Report
Comments and Suggestions for Authors
Peer Review Comments
Overall Assessment
This study's title successfully conveys its focus on promoting One Health concepts in Bulgaria since it is concise and well-structured. It places emphasis on the application of a mixed-methods approach, which proposes an all-encompassing methodology that blends quantitative and qualitative methodologies. The inclusion of "transdisciplinary research and practice" highlights the study's integrative and collaborative nature, which is consistent with the essential principles of the One Health concept. Overall, the title is interesting and well communicates the breadth and creative methodology of the study.
Introduction section
The introduction section have some missing information. The introduction section should clearly the rationale of this study and how the one health (OH) is key in this study. For instance, the introduction section should interact with the following questions such as:
- In tackling today's public health issues, what is the One Health (OH) concept's regional or worldwide significance? In what ways does this study support ongoing national or international initiatives to apply the ideas of One Health? What particular issues or problems does this initiative want to address in the areas of antimicrobial resistance (AMR) and public health?
- Why does Bulgaria represent a significant case study for the advancement of One Health principles?
- Clearly state the overall objective and the specific objectives of this study. In contrast to other OH initiatives, what special contribution does this project offer? How do the report's primary goals relate to the research project's broader framework?
Method section
In the method section, the Authors rightly employed the appropriate method such as the Nominal Group Technique (NGT) for this study. This method is a structured decision-making method that encourages equal participation by having individuals generate, share, and discuss ideas independently before prioritizing them through voting. It fosters collaboration, minimizes bias, and produces well-considered group decisions. However, there was no where in this study where Nominal group Technique (NGT) was conducted in this study. The Authors should clearly explain how they conducted the NGT among the different participants. This is very key and the Authors should clearly address this concern.
The Authors did not employ the use of parallel mixed-methods design in this study. What is done in this study is only quantitative (the use of structured questionnaire). There was no qualitative aspect mentioned in the method section. The nominal group technique (NGT) should have provided the qualitative section for this study. The Authors should know that the parallel mixed-methods design, also known as the convergent parallel design, is a flexible research approach, involving simultaneous but independent collection and analysis of quantitative and qualitative data during the same phase of the study. The results from both methods are then integrated, providing a comprehensive understanding of the research problem. This approach is particularly effective for addressing complex questions, as it allows for the comparison, corroboration, or combination of findings, demonstrating its adaptability to different research questions.
The Authors should make the Figure 1 bigger and clearer for your Readers to see the framework adopted or adapted in this study.
Reference Figure 1, I am very sure Figure 1 was adopted or adapted from a broader framework. Kindly cite the author’s work for this Figure 1.
There should be a statistical method section which shows the type of analysis was carried out by each of the specific objective of this study. The analytical method should be clearly spelt out.
What is the ethical consideration employed in this study? What were the guiding ethical principles employed in this study, in terms of confidentiality, beneficence, non-maleficence etc. How did you provide the participant informed consent form as well as participant information sheet? Describe how the ethical process was carried out during the data collection?
Result section
First, the result section should begin with the socio-demographic characteristics of the respondents and the quantitative data can provide the respondents’ demographic characteristics.
Second, the result section should start with presentation of the quantitative data findings, along with a concise interpretation of the findings for each objective as a theme.
Third, there was no qualitative data findings in this section. You mentioned parallel mixed-methods but it is only the quantitative data that you have reported in the result section. Why? You see that my comments dwell more that your method section is faulty. You need to correct them all in order to have a better result section.
The key findings are nt presented with sufficient clarity and they are not accompanied by the necessary statistical details to support the interpretations for this study. The narrative of the results do not align well with the study as the specific objectives are not stated in the introduction section.
Discussion section
The Authors have not effectively connected the study's findings to the broader literature and did not highlight their implications for any theory, practice, or policy.
The discussion section must integrate existing knowledge, as this will emphasize the unique contribution of this study and provide a comprehensive understanding of the topic. The discussion section should be built along with result section.
Strength and Limitations of the study
The authors have not adequately acknowledged the study's strength and limitations of this study. Kindly provide the information on the strengths and limitations for this study.
1) What methodological strengths, such as sample size, data collection techniques, and analysis methods, enhance the reliability and validity of the study's findings?
2) What are the potential limitations of the study, including sample representativeness, response biases, and other confounding factors, and how might these limitations affect the interpretation and generalizability of the results? These key questions should be interacted with and helps to provide this section with the core information.
Future research directions
The authors have not adequately proposed meaningful directions for future research replications, which is a key responsibility in advancing the field. This demonstrates critical reflection and paves the way for further exploration in different settings.
Conclusion/Recommendation
This section should stem from the Result section, and recommendations should be provided for this study. The recommendations could be in form of policy, practice and research interventions.
Referencing and in-citations
Referencing and in-citations in your study - Double-check all in-citations and references to ensure they are correctly cited and match the sources listed in the reference section. Include the most recent and relevant sources to ensure that your study reflects the latest research developments in the field. Use a consistent citation style (e.g., APA, MLA) throughout the manuscript, ensuring all references are formatted correctly according to the chosen style. Ensure all references are from reputable and peer-reviewed journals, academic publications, or authoritative sources. Compare cited information with other credible sources to confirm its accuracy and reliability. Arrange references alphabetically by the author's last name and ensure they are easily located within the reference section.
Proofreading and editing
Engaging a professional editor can significantly enhance your manuscript. Editors improve structure, content clarity, language, and style while ensuring proper formatting and consistency. They also verify references, provide constructive feedback, and perform thorough proofreading, increasing the manuscript's quality and chances of publication.
Comments on the Quality of English LanguagePeer Review Comments
Overall Assessment
This study's title successfully conveys its focus on promoting One Health concepts in Bulgaria since it is concise and well-structured. It places emphasis on the application of a mixed-methods approach, which proposes an all-encompassing methodology that blends quantitative and qualitative methodologies. The inclusion of "transdisciplinary research and practice" highlights the study's integrative and collaborative nature, which is consistent with the essential principles of the One Health concept. Overall, the title is interesting and well communicates the breadth and creative methodology of the study.
Introduction section
The introduction section have some missing information. The introduction section should clearly the rationale of this study and how the one health (OH) is key in this study. For instance, the introduction section should interact with the following questions such as:
- In tackling today's public health issues, what is the One Health (OH) concept's regional or worldwide significance? In what ways does this study support ongoing national or international initiatives to apply the ideas of One Health? What particular issues or problems does this initiative want to address in the areas of antimicrobial resistance (AMR) and public health?
- Why does Bulgaria represent a significant case study for the advancement of One Health principles?
- Clearly state the overall objective and the specific objectives of this study. In contrast to other OH initiatives, what special contribution does this project offer? How do the report's primary goals relate to the research project's broader framework?
Method section
In the method section, the Authors rightly employed the appropriate method such as the Nominal Group Technique (NGT) for this study. This method is a structured decision-making method that encourages equal participation by having individuals generate, share, and discuss ideas independently before prioritizing them through voting. It fosters collaboration, minimizes bias, and produces well-considered group decisions. However, there was no where in this study where Nominal group Technique (NGT) was conducted in this study. The Authors should clearly explain how they conducted the NGT among the different participants. This is very key and the Authors should clearly address this concern.
The Authors did not employ the use of parallel mixed-methods design in this study. What is done in this study is only quantitative (the use of structured questionnaire). There was no qualitative aspect mentioned in the method section. The nominal group technique (NGT) should have provided the qualitative section for this study. The Authors should know that the parallel mixed-methods design, also known as the convergent parallel design, is a flexible research approach, involving simultaneous but independent collection and analysis of quantitative and qualitative data during the same phase of the study. The results from both methods are then integrated, providing a comprehensive understanding of the research problem. This approach is particularly effective for addressing complex questions, as it allows for the comparison, corroboration, or combination of findings, demonstrating its adaptability to different research questions.
The Authors should make the Figure 1 bigger and clearer for your Readers to see the framework adopted or adapted in this study.
Reference Figure 1, I am very sure Figure 1 was adopted or adapted from a broader framework. Kindly cite the author’s work for this Figure 1.
There should be a statistical method section which shows the type of analysis was carried out by each of the specific objective of this study. The analytical method should be clearly spelt out.
What is the ethical consideration employed in this study? What were the guiding ethical principles employed in this study, in terms of confidentiality, beneficence, non-maleficence etc. How did you provide the participant informed consent form as well as participant information sheet? Describe how the ethical process was carried out during the data collection?
Result section
First, the result section should begin with the socio-demographic characteristics of the respondents and the quantitative data can provide the respondents’ demographic characteristics.
Second, the result section should start with presentation of the quantitative data findings, along with a concise interpretation of the findings for each objective as a theme.
Third, there was no qualitative data findings in this section. You mentioned parallel mixed-methods but it is only the quantitative data that you have reported in the result section. Why? You see that my comments dwell more that your method section is faulty. You need to correct them all in order to have a better result section.
The key findings are nt presented with sufficient clarity and they are not accompanied by the necessary statistical details to support the interpretations for this study. The narrative of the results do not align well with the study as the specific objectives are not stated in the introduction section.
Discussion section
The Authors have not effectively connected the study's findings to the broader literature and did not highlight their implications for any theory, practice, or policy.
The discussion section must integrate existing knowledge, as this will emphasize the unique contribution of this study and provide a comprehensive understanding of the topic. The discussion section should be built along with result section.
Strength and Limitations of the study
The authors have not adequately acknowledged the study's strength and limitations of this study. Kindly provide the information on the strengths and limitations for this study.
1) What methodological strengths, such as sample size, data collection techniques, and analysis methods, enhance the reliability and validity of the study's findings?
2) What are the potential limitations of the study, including sample representativeness, response biases, and other confounding factors, and how might these limitations affect the interpretation and generalizability of the results? These key questions should be interacted with and helps to provide this section with the core information.
Future research directions
The authors have not adequately proposed meaningful directions for future research replications, which is a key responsibility in advancing the field. This demonstrates critical reflection and paves the way for further exploration in different settings.
Conclusion/Recommendation
This section should stem from the Result section, and recommendations should be provided for this study. The recommendations could be in form of policy, practice and research interventions.
Referencing and in-citations
Referencing and in-citations in your study - Double-check all in-citations and references to ensure they are correctly cited and match the sources listed in the reference section. Include the most recent and relevant sources to ensure that your study reflects the latest research developments in the field. Use a consistent citation style (e.g., APA, MLA) throughout the manuscript, ensuring all references are formatted correctly according to the chosen style. Ensure all references are from reputable and peer-reviewed journals, academic publications, or authoritative sources. Compare cited information with other credible sources to confirm its accuracy and reliability. Arrange references alphabetically by the author's last name and ensure they are easily located within the reference section.
Proofreading and editing
Engaging a professional editor can significantly enhance your manuscript. Editors improve structure, content clarity, language, and style while ensuring proper formatting and consistency. They also verify references, provide constructive feedback, and perform thorough proofreading, increasing the manuscript's quality and chances of publication.
Reviewer 2 Report
Comments and Suggestions for Authors
The manuscript titled "Advancing One Health Principles in Bulgaria: A Mixed-Methods Approach to Transdisciplinary Research and Practice" by Vankova et al., is important in its area of study for developing One Health strategies to curb spread of resistance. However, the following comments need be addressed:
Title
The study's specific aims and methodology is not reflected in the title which is broad. E.g., the phrase "Advancing One Health Principles" is vague, one cannot comprehend whether it means implementation, awareness, or policy development. To clarify the focus on strategy implementation and policy, I suggest the title be changed to "Implementation of One Health Strategies in Bulgaria: A Parallel Mixed-Methods Approach to Transdisciplinary Research and Policy Development."
Abstract
There was no specific thing regarding methodology and key findings. General terms such as "raising awareness" and "promoting OH principles" were overused without defining specific measurable outcomes. This phrase "structured into three chapters" is awkward, it would be better the authors refer to study components. Sample size or key quantitative results were not indicated. The authors should add a concise description of the study design and sample. They should also use "divided into three thematic research areas: OH & AMR, OH & Wellbeing, and OH Education & Dissemination." Rather than "structured into three chapters". At least one major quantitative result should be added to strengthen the abstract.
Introduction
37–43: The research justification is not clear and is overly philosophical
44: "Following the COVID-19 crisis" Authors should specify the relevance of COVID-19 to this study.
45: "NACID Registry 2412381, â„– 22007" Please provide a brief explanation of this reference
44–49: Redundant explanations of One Health (OH). It is better to integrate this within the literature review
55–63: The authors claim that OH serves as an "overarching framework" for various research areas without sufficient supporting evidence
The authors should begin with a clear problem statement that shows the critical challenges faced by Bulgaria that is necessitating an integrated One Health (OH) approach." A more structured rationale for the study should be made instead of a broad conceptual discussion.
Materials and Methods
Issues:
71–75: The parallel mixed-methods design (PMMD) is description is not clear.
Authors should clearly describe how surveys were distributed and the response rate
84–86: Ethics approval should be placed in a dedicated section titled “Ethical Considerations” The justification for using the nominal group technique, is missing. The PMMD approach should be clearly defined and the reason why it was chosen. The number of participants per method (survey, focus groups, etc.) should be specified
168–170: This phrasing is awkward and should be revised "content analysis of national policy documents revealed significant gaps, particularly the absence of a comprehensive OH Action Plan."
Results
Inconsistent data presentation in Table 1 wherein some parts use percentages while others only list numbers. In the KAP survey, what are the implications of low awareness of OH principles? I suggest the authors add statistical comparisons to show significant findings, e.g., knowledge gaps between medical and veterinary professionals
181–190: Interpretation of the survey data is insufficient
The authors claim "policy gaps" were identified, but they did not make any specific policy recommendations. Discussion on the impact of these findings, should be expanded, that is what do they mean for Bulgarian public health policy?
Discussion
There is no strong link between findings and recommendations. Specific policy recommendations based on findings should be discussed
234–242: The discussion on terminology translation is unnecessary in a scientific study. Focus on scientific implications and remove unnecessary linguistic discussion
250–254: Since other approaches exist, it is too absolute to say that OH is "the only possible frame to tackle AMR" The authors should use phrases such as "OH represents a promising framework" instead of "the only possible solution."
Conclusion
Too general/broad. It lacked specificity regarding study impact. Please summarize key findings concisely
253–254: This phrase "bolster resilience in public health systems" should be backed by a suggested concrete actions otherwise it is vague. Provide specific recommendations e.g., "Integrating OH into Bulgarian medical curricula and policy frameworks could improve national AMR strategies."
Reviewer 3 Report
Comments and Suggestions for Authors
Comments to the manuscript; ijerph-3511026
Advancing One Health Principles in Bulgaria: A Mixed-Methods Approach to Transdisciplinary Research and Practice
Specific comments to the manuscript:
Abstract
The abstract needs to be re-written, as much of methodology and less results (findings) are included. Please maintain the flow of sentence either in past tense or present (is/are/was). Reduce the methodology and write brief findings. Additionally, remove the number/point numbers indentation and make a paragraph of the points (Line 22-27).
Introduction:
Line 39: balanced_ healthy …..balanced and healthy
Line 40: foundational principles…..foundation of
Line 43: walls between humanities…….walls between and human and
Line 47: project has been to raise…….project was to
Line 53-53: Therefore, the long-term project ambition grew to new heights, viewing OH as a social phenomenon that has the potential to unite, introducing the OH idea into the broader public health context….Either restructure or remove the sentence as meaning and words are repetitive of what has already been sent in the text.
Line 64-66: Happily, the convergent potential of the OH concept served as an overarching framework, bringing the team together under a shared commitment to scientific advancements in service of society…..either restructure or remove the sentence.
General comment:
Table 1: Why only 60 Veterinarian as comparison to 168 medical professionals..? Any reason—was it less number of veterinarians available or any other
Provide with the detailed results of KAP survey. The results provided are not sufficient. There are various gaps disrupting the flow of manuscript, which needs to be addressed. Language/grammar checks are needed. Discussion needs to be elaborated and most importantly results needs to be re-written in terms of questions/items of questionnaire, responses, and statistical analysis performed in the form of table.
Comments on the Quality of English Language
Language needs improvement
Round 2
Reviewer 1 Report
Comments and Suggestions for Authors
The authors have thoroughly addressed all the comments raised during the review process. They have provided clear and comprehensive responses, incorporating the necessary revisions to enhance the quality and clarity of the manuscript. Having carefully reviewed the revised submission, I am confident that it meets the journal's standards for publication. Therefore, I wholeheartedly recommend the manuscript for publication.
Comments on the Quality of English LanguageThe manuscript is written in clear and concise English, using appropriate technical terminology and academic language. The structure and flow of the text are logical and coherent, ensuring that the intended message is effectively conveyed. While the overall quality of the English is high, minor typographical or grammatical adjustments may further enhance clarity and readability.
Reviewer 2 Report
Comments and Suggestions for Authors
The manuscript have been improved significantly
Reviewer 3 Report
Comments and Suggestions for Authors
No further revisions needed. All my queries have been addressed.